# Integrated Osteoporosis Care to Reduce Denosumab-Associated Hypocalcemia for Patients with Advanced Chronic Kidney Disease and End-Stage Renal Disease

**DOI:** 10.3390/healthcare11030313

**Published:** 2023-01-20

**Authors:** Chia-Tien Hsu, Ya-Lian Deng, Mu-Chi Chung, Shang-Feng Tsai, Shih-Yi Lin, Cheng-Hsu Chen

**Affiliations:** 1Division of Nephrology, Department of Internal Medicine, Taichung Veterans General Hospital, Taichung 407219, Taiwan; 2School of Medicine, National Yang Ming Chiao Tung University, Taipei 112304, Taiwan; 3Center for Osteoporosis Prevention and Treatment, Taichung Veterans General Hospital, Taichung 407219, Taiwan; 4Department of Post-Baccalaureate Medicine, College of Medicine, National Chung Hsing University, Taichung 40227, Taiwan

**Keywords:** denosumab-associated hypocalcemia, advanced chronic kidney disease, end-stage renal disease, integrated osteoporosis care

## Abstract

The incidence of hypocalcemia is high in patients with chronic kidney disease (CKD) and end-stage renal disease (ESRD) undergoing denosumab treatment. Since 2018, we have carried out a “multidisciplinary integrated care program for osteoporosis among patients with CKD and ESRD” in our hospital. The aim of this study was to compare the incidence of denosumab-associated hypocalcemia among patients with advanced CKD and ESRD before and after the integrated care program. We retrospectively reviewed the records of patients on their first dose of denosumab treatment from January 2012 to December 2021. A total of 3208 patients were included in our study. Among the 3208 patients, there were 101 dialysis patients, 150 patients with advanced CKD (stage 4 and 5), and 2957 patients with an estimated glomerular filtration rate (eGFR) higher than or equal to 30. The incidence of post-treatment severe hypocalcemia (corrected calcium level less than 7.0 mg/dl) within 30 days was significantly higher in the dialysis and advanced CKD group than in patients with an eGFR higher than or equal to 30 (6.9% vs. 2.0% vs. 0.1%, respectively, *p* < 0.001). Based on the results of the multivariate regression model, poor renal function (*p* < 0.05) and lower baseline corrected calcium level (*p* < 0.05) were associated with severe hypocalcemia within 30 days following the first dose of denosumab treatment. The incidence of post-treatment severe hypocalcemia within 30 days in advanced CKD and dialysis patients was significantly lower after the integrated care program (6.8% vs. 0.8%, *p* < 0.05). Our study shows that multidisciplinary integrated care may reduce the incidence rate of denosumab-associated severe hypocalcemia among patients with advanced CKD and ESRD.

## 1. Introduction

Chronic kidney disease–mineral bone disease (CKD-MBD) is one of the most common complications among patients with chronic kidney disease (CKD) and affects the majority of patients with advanced CKD and end-stage renal disease (ESRD) [1]. The skeletal derangements of CKD-MBD are associated with a higher risk of osteoporosis and fractures. According to previous studies, patients with CKD have a more than 2.5-fold higher risk of fractures, and those with ESRD have a more than 4-fold risk compared to people who do not have CKD [2,3,4]. Fragility fractures are related to higher mortality and hospitalization costs in patients with advanced CKD and ESRD [5]. Therefore, attention should be paid to this issue.

According to previous studies [6,7,8,9,10,11,12,13], the incidence of hypocalcemia is higher in patients with advanced CKD and ESRD undergoing denosumab treatment than in patients not administered denosumab. There are multiple mechanisms related to denosumab-associated hypocalcemia among advanced CKD and dialysis patients [8,10]. As renal function declines, the laboratory abnormalities related to CKD-MBD worsen. Hyperphosphatemia, hyperparathyroidism, high levels of fibroblast growth factor-23 (FGF-23), low levels of 1,25-dihydroxyvitamin D (calcitriol), and hypocalcemia characterize patients with advanced CKD and ESRD [14,15]. Denosumab is a monoclonal antibody against the receptor activator of nuclear factor kappa B ligand (RANK-L), which interferes with osteoclast formation, activity, and survival. Consequently, it reduces bone resorption and increases the bone mineral density [16]. As denosumab is not cleared by the kidneys, it has been increasingly used in patients with advanced CKD and ESRD [8,17,18,19]. Denosumab downregulates osteoclast activity, reduces bone resorption, and further decreases the release of calcium from the bone, further increasing the risk of hypocalcemia among patients with advanced CKD and dialysis patients [10].

Some approaches to reducing denosumab-associated hypocalcemia have been advocated for by specialists in osteoporosis and kidney disease, such as (1) treating CKD-MBD before denosumab use; (2) careful patient selection; (3) checking baseline calcium and correcting hypocalcemia prior to denosumab use; (4) calcium and vitamin D supplementation during denosumab treatment; (5) increasing the dialysate calcium concentration during denosumab treatment in dialysis patients; and (6) closely monitoring the calcium levels after denosumab treatment [8,10,18]. Since 2018, we have been carrying out a “multidisciplinary integrated care program for osteoporosis among patients with CKD and ESRD” in our hospital. We have implemented many active measures to reduce denosumab-associated hypocalcemia.

Mild hypocalcemia may be asymptomatic or cause muscle cramps, but severe hypocalcemia can be life-threatening [20]. Severe hypocalcemia is highly associated with patient safety, so we focused on the incidence rate of denosumab-associated severe hypocalcemia in this study. In addition, our integrated care program is mainly concentrated on the prevention of severe hypocalcemia in patients with advanced CKD and ESRD undergoing denosumab treatment. The aim of this study was to compare the incidence rate of denosumab-associated severe hypocalcemia among patients with CKD and ESRD before and after the implementation of our integrated care program.

## 2. Materials and Methods

### 2.1. Integrated Osteoporosis Care Program among Patients with Advanced CKD and ESRD

The incidence of hypocalcemia is higher in patients with advanced CKD and ESRD undergoing denosumab treatment than in patients not administered denosumab, so our integrated care program focused on patients with advanced CKD and ESRD. We built a multidisciplinary integrated care program for osteoporosis among patients with advanced CKD and ESRD from 2018 onwards. The team members involved include nephrologists, orthopedists, osteoporosis-qualified nursing consultants (case managers), dialysis nurses, and dieticians.

We applied the computerized physician order entry (CPOE) system [21] in our electronic health information system (EHIS) and set up a CPOE module for the osteoporosis integrated care program among patients with advanced CKD and ESRD. This module comprised the orders of laboratory examinations, dual-energy X-ray absorptiometry (DEXA) examination, and a referral sheet for osteoporosis-qualified nursing consultants (case managers). The laboratory examinations included renal function, calcium, phosphorus, albumin, alkaline phosphatase, intact parathyroid hormone, and other endocrine profiles related to secondary osteoporosis.

Osteoporosis-qualified nursing consultants (case managers) further referred patients with advanced CKD and ESRD to nephrologists for the evaluation of CKD-MBD and patient selection before denosumab treatment. According to the European consensus statement on the diagnosis and management of osteoporosis in patients with advanced CKD and ESRD [22], CKD-MBD therapy should be optimized before denosumab treatment. The risks and benefits of denosumab treatment need to be balanced and discussed with the patient. Nephrologists are responsible for the optimizing of CKD-MBD therapy and the risks and benefits judgement before denosumab treatment. Denosumab may worsen the bone structure and vascular calcification among CKD patients with low-turnover bone disease. We excluded patients with low levels of parathyroid hormone and bone metabolic markers from denosumab treatment.

In addition, we also set up a warning system in our EHIS to ensure that the patients undergoing denosumab treatment had (1) been checked for baseline calcium level and corrected hypocalcemia prior to denosumab use; (2) received calcium (3 g/day) and vitamin D (0.5 to 2 μg/day) supplements during denosumab treatment [18]; and (3) been monitored regarding their calcium levels after denosumab treatment. All of the dialysis patients in our hospital received calcium and vitamin D supplements during denosumab treatment. Moreover, we also prophylactically increased the dialysate calcium concentration during denosumab treatment for both hemodialysis and peritoneal dialysis patients. After that, we checked the calcium and phosphate levels every week in the hemodialysis patients for one month and every two weeks in the peritoneal dialysis patients for one month [8,10,18].

### 2.2. Study Design and Subjects

We retrospectively reviewed 10 years of medical records of patients on their first dose of denosumab treatment from January 2012 to December 2021 in our hospital. Our study was approved by the institutional review board of Taichung Veterans General Hospital (IRB TCVGH No: CE22167A). Patient informed consent was waived due to the retrospective data analysis nature of this study. We excluded patients without baseline laboratory data before denosumab treatment, without dual-energy X-ray absorptiometry (DEXA) examination, or without a diagnosis code for osteoporosis. The baseline laboratory data were defined as the last laboratory data within three months prior to denosumab treatment.

Hypocalcemia was defined as a corrected calcium level less than 8.5 mg/dl. Mild hypocalcemia was defined as a corrected calcium level higher than or equal to 7.0 mg/dl and less than 8.5 mg/dl (7.0 ≦ corrected calcium < 8.5 mg/dl). Severe hypocalcemia was defined as a corrected calcium level less than 7.0 mg/dl. We analyzed the incidence of post-treatment hypocalcemia within 30 days following the first dose of denosumab treatment among different renal function groups to confirm the target patient group of the integrated care program. Then, we analyzed the incidence of severe hypocalcemia within 30 days following the first dose of denosumab treatment before and after the implementation of the integrated care program, especially in the subgroup with advanced CKD (eGFR less than 30) and ESRD.

A total of 3208 patients on their first dose of denosumab treatment were included in our study. Based on the baseline renal function, the 3208 patients on their first dose of denosumab treatment were divided into three groups: patients with an estimated glomerular filtration rate (eGFR) higher than or equal to 30 (group 1, n = 2957); patients with CKD stage 4 and 5 (group 2, n = 150); and dialysis patients (group 3, n = 101). We compared the incidence of post-treatment hypocalcemia within 30 days of the first dose of denosumab in the different subgroups, especially in the subgroup before and after the integrated care program. Because the half-life of denosumab is around 30 days [23] and clinical hypocalcemia tends to occur within 7–20 days following the first dose of denosumab [6,8,9,10,11,17,18,19], we monitored hypocalcemia within 30 days following the first dose of denosumab as our outcome (calcium level monitoring followed the protocol shown in Section 2.1).

In order to compare the incidence rate of denosumab-associated severe hypocalcemia among patients with advanced CKD and ESRD (the target group of our integrated care program) before and after the implementation of the integrated care program, the patients with advanced CKD and ESRD were divided into the following two groups: (1) those who were treated from January 2012 to December 2017 as the dataset of “before integrated care”; and (2) those who were treated from January 2018 to December 2021 as the set of “after integrated care”.

### 2.3. Statistical Analysis

The continuous variables with normal distribution are shown as mean ± standard deviation, whereas the continuous variables with non-normal distribution are presented as the median (first quartile, third quartile). The assessment of normality was conducted using the Kolmogorov–Smirnov test and Shapiro–Wilk test. The categorical variables are reported as numbers (percentage). Tests for the statistical significance were conducted using the Fisher’s exact test or the Chi-squared test for categorical variables and the Wilcoxon signed-rank test for non-parametric continuous variables of paired data. For multiple comparisons, the Kruskal–Wallis test was used for non-parametric variables. We used the Cox proportional hazards regression model to analyze factors that may be associated with severe hypocalcemia within 30 days following the first dose of denosumab treatment. Variables in the univariate analysis with a *p* value < 0.1 were considered as potential confounders in the multivariate models. Multivariate analysis was performed to adjust for the confounders selected in the univariate analysis. The level of significance was set at *p* < 0.05. Statistical analyses were performed using MedCalc for Windows, version 20.114 (MedCalc Software, Ostend, Belgium).

## 3. Results

### 3.1. Comparing the Incidence of Post-Treatment Hypocalcemia within 30 Days among Different Renal Function Groups on Denosumab

A total of 3208 patients on their first dose of denosumab treatment were included in our study. There were 602 male and 3202 female patients, with a median age of 73.7 years. Among the 3208 patients, there were 101 dialysis patients, 150 patients with advanced CKD (stage 4 and 5), and 2957 patients with an eGFR higher than or equal to 30. The average day of hypocalcemia testing was 13.0 ± 7.2 days.

Table 1 shows the clinical characteristics of the patients that received their first dose of denosumab treatment and the comparison of the three renal function groups. Due to the influence of CKD-MBD, the age of the patients undergoing denosumab treatment was significantly lower in the dialysis group. The percentage of patients with co-morbidities was also significantly higher in the dialysis and advanced CKD groups. The median estimated glomerular filtration rate (eGFR) in group 1 was 78.3 (62.4–94.3) mL/min/1.73 m^2^, while the eGFR in group 2 was 21.9 (14.1–26.4) mL/min/1.73 m^2^. Compared with group 1 (eGFR ≧ 30), the dialysis group had higher corrected calcium level (9.10 vs. 9.35; *p* < 0.001), higher serum phosphate level (3.6 vs. 4.4; *p* < 0.001), higher alkaline phosphatase level (103.0 vs. 135.5; *p* < 0.001), and higher intact parathyroid hormone level (52.3 vs. 341.9; *p* < 0.001) before denosumab treatment.

Although the corrected calcium level was higher in the dialysis group, Figure 1 reveals that the incidence of hypocalcemia within 30 days post treatment was significantly higher in the dialysis and advanced CKD groups than in patients with an eGFR higher than or equal to 30 (17.8% vs. 12.7% vs. 2.4%, respectively, *p* < 0.001). 

Similarly, the rates of “severe” hypocalcemia within 30 days were significantly higher in the dialysis and advanced CKD groups than in patients with eGFR higher than or equal to 30 (6.9% vs. 2.0% vs. 0.1%, respectively, *p* < 0.001). Our results are similar to those of previous studies—the incidence of hypocalcemia is higher in patients with advanced CKD and ESRD undergoing denosumab treatment. Hence, our integrated care program focused on patients with advanced CKD and ESRD.

We used the Cox proportional hazards regression model to analyze factors that may be associated with “severe” hypocalcemia within 30 days following the first dose of denosumab treatment. The Cox proportional hazards model for severe hypocalcemia within 30 days following the first dose of denosumab treatment is summarized in Table 2. In the univariate regression, poor renal function (*p* < 0.01), younger age (*p* < 0.01), lower BMI (*p* < 0.05), patients with hypertension (*p* < 0.05), lower baseline corrected calcium level (*p* < 0.01), higher baseline phosphate level (*p* < 0.01), higher baseline alkaline phosphatase level (*p* < 0.01), and higher baseline intact parathyroid hormone (*p* < 0.01) were associated with severe hypocalcemia within 30 days of the first dose of denosumab treatment. Patients with integrated care demonstrated a reduced risk of severe hypocalcemia within 30 days following the first dose of denosumab treatment as compared with patients without integrated care (before the implementation of integrated care). Based on the results of the multivariate regression model, poor renal function (*p* < 0.05) and lower baseline corrected calcium level (*p* < 0.05) were associated with severe hypocalcemia within 30 days following the first dose of denosumab treatment.

### 3.2. The Incidence of Hypocalcemia within 30 Days Following the First Dose of Denosumab Treatment before and after the Implementation of the Integrated Care Program

We compared the incidence of hypocalcemia within 30 days following the first dose of denosumab treatment before and after the implementation of the integrated care program. Figure 2a shows that the incidence of hypocalcemia within 30 days post treatment in patients with an eGFR higher than or equal to 30 after the implementation of the integrated care program was not statistically different compared to that before the start of the program. However, Figure 2b reveals that the incidence of any hypocalcemia within 30 days post treatment in advanced CKD and dialysis patients was significantly lower after the integrated care program was implemented (18.9% vs. 10.1%, *p* < 0.05). The incidence of severe hypocalcemia within 30 days post treatment in advanced CKD and dialysis patients was significantly lower after the integrated care program was implemented (6.8% vs. 0.8%, *p* < 0.05).

Because our integrated care program focused on patients with advanced CKD and ESRD, unsurprisingly, our integrated care program did not reduce the incidence of hypocalcemia within 30 days following the first dose of denosumab treatment among patients with an eGFR higher than or equal to 30. Hence, we further analyzed the clinical characteristics before and after integrated care among patients with advanced CKD and ESRD. Severe hypocalcemia is highly associated with patient safety, so we focused on the incidence rate of denosumab-associated severe hypocalcemia in this study.

Table 3 shows the comparison of the clinical characteristics before and after the integrated care program among patients with advanced CKD and ESRD (the target group of our integrated care program). The demographic characteristics, percentage of the renal function subgroup, percentage of comorbidities, and baseline laboratory data after the integrated care program were not statistically different compared to before the program. However, the incidence of hypocalcemia within 30 days post denosumab treatment in advanced CKD and dialysis patients was significantly lower after the integrated care program.

The Cox proportional hazards model for severe hypocalcemia within 30 days following the first dose of denosumab treatment among patients with advanced CKD and ESRD (the target group of our integrated care program) is summarized in Table 4. In the univariate regression, patients without integrated care (*p* < 0.05) and younger (*p* < 0.01) were associated with severe hypocalcemia within 30 days of the administration of first dose of denosumab. Patients with integrated care demonstrated a reduced risk of severe hypocalcemia within 30 days following the first dose of denosumab treatment as compared with patients without integrated care (before the implementation of the integrated care). However, no significant factors were associated with severe hypocalcemia within 30 days following the first dose of denosumab treatment based on the results of the multivariate regression model.

Figure 3 presents the incidence within 30 days post denosumab treatment of hypocalcemia before and after the integrated care program was implemented among “dialysis” patients in our hospital. After the integrated care program was implemented, the incidence of hypocalcemia within 30 days post treatment among the dialysis patients decreased from 24.5% to 11.5% in our hospital. The incidence of severe hypocalcemia within 30 days post treatment among the dialysis patients was significantly lower after the integrated care program was implemented (12.2% vs. 1.9%, *p* < 0.05).

## 4. Discussion

Our study confirmed that the target patient group of our integrated care program is patients with advanced CKD and ESRD. In addition, we also showed the benefit of this integrated care program among the target patient group. Patients with advanced CKD and ESRD often have CKD-MBD, including vascular or soft tissue calcification, secondary hyperparathyroidism, and renal osteodystrophy [1,24]. Our study showed patients on denosumab with poor renal function were younger, owing to the CKD-MBD-related secondary osteoporosis. In the univariate regression model, younger patients were associated with severe hypocalcemia. This may be related to these younger patients mainly being dialysis patients.

To our knowledge, few studies compared the incidence of severe hypocalcemia after approaches to reducing denosumab-associated hypocalcemia. A poor renal function is a stronger factor for the development of severe hypocalcemia, as shown in Table 2. However, Table 4 reveals that a poor renal function was not a significant risk factor for severe hypocalcemia. This finding is similar to that of the study reported by Saito et al. [25]. They reported that renal impairment is not a risk factor for denosumab-induced hypocalcemia in a strict denosumab administration management system with calcium and vitamin D supplementation. After the implementation of a management system with calcium and vitamin D supplementation, the risk of denosumab-induced hypocalcemia among patients with poor renal function was reduced. However, there were only two patients with an eGFR less than 30 in their study. Our study further confirms the benefit of a management system (or integrated care program) with calcium and vitamin D supplementation, decreasing the risk of denosumab-induced hypocalcemia among patients with advanced CKD and ESRD.

In addition, we also prophylactically increased the dialysate calcium concentration during denosumab treatment for both hemodialysis and peritoneal dialysis patients. According to previous studies [7,8,9,17,18,19], the incidence of hypocalcemia after denosumab treatment in dialysis patients ranges from 25.5% to 63%. Thongprayoon et al. [12] conducted a meta-analysis of observational studies in 2018 and reported that the pooled estimated incidence of hypocalcemia following denosumab treatment in ESRD patients was 42%. The incidence of hypocalcemia within 30 days post treatment among ESRD patients was 24.5% before the integrated care program was implemented in our hospital, as shown in Figure 3. This incidence is similar to that reported in a study by Hiramatsu et al. (25.5%) [9]. After the integrated care program was implemented, the incidence of hypocalcemia within 30 days post treatment among ESRD patients decreased from 24.5% to 11.5% in our hospital. Our study showed that prophylactically increasing the dialysate calcium concentration with calcium and vitamin D supplementation during denosumab treatment in both hemodialysis and peritoneal dialysis patients could reduce hypocalcemia complications caused by denosumab in dialysis patients.

Our results showed no significant improvement of mild hypocalcemia after the implementation of the integrated care program. This may be caused by different doctors’ compliance with the integrated care program for the dose adjustment of calcium and vitamin D. Because mild hypocalcemia may be asymptomatic, some doctor may keep the patient under observation. Our integrated care program also included a warning system to ensure that the patients undergoing denosumab treatment had (1) been checked for baseline calcium level and corrected hypocalcemia prior to denosumab use; (2) received a calcium (3 g/day) and vitamin D (0.5 to 2 μg/day) supplement during denosumab treatment; (3) been monitored regarding their calcium levels after denosumab treatment. The warning system may increase the doctors’ and healthcare professionals’ awareness of denosumab-associated hypocalcemia.

Some limitations of the current study should be acknowledged. First, the retrospective nature of the study may have led to some unrecognized confounding factors to bias the findings. Second, some patients with asymptomatic hypocalcemia could not be identified because laboratory monitoring was not undertaken, especially among patients treated before the integrated care program was implemented and patients without dialysis treatment. The frequency of serum calcium monitoring was significantly higher in dialysis patients than in patients without dialysis treatment, which may bias the episodes of hypocalcemia. The higher incidence of hypocalcemia among dialysis patients may be due to monitoring timing variation between groups in our study. Third, the study population had a selection bias for one location. The case number of advanced CKD and ESRD in this study was relatively small because this study was a single-center retrospective study. Further studies with a larger sample size are required to confirm the efficacy of these approaches for reducing hypocalcemia complications.

## 5. Conclusions

The incidence of post-treatment hypocalcemia within 30 days following the first dose of denosumab is higher in patients with advanced CKD and ESRD as compared to patients with an eGFR higher than or equal to 30. Doctors’ and healthcare professionals’ awareness of denosumab-associated hypocalcemia and the implementation of approaches to reduce this complication in patients with advanced CKD and ESRD are important considerations for patient safety. Our study showed that multidisciplinary integrated care may reduce the incidence rate of denosumab-associated severe hypocalcemia among patients with advanced CKD and ESRD, thereby improving patient safety.

## Figures and Tables

**Figure 1 healthcare-11-00313-f001:**
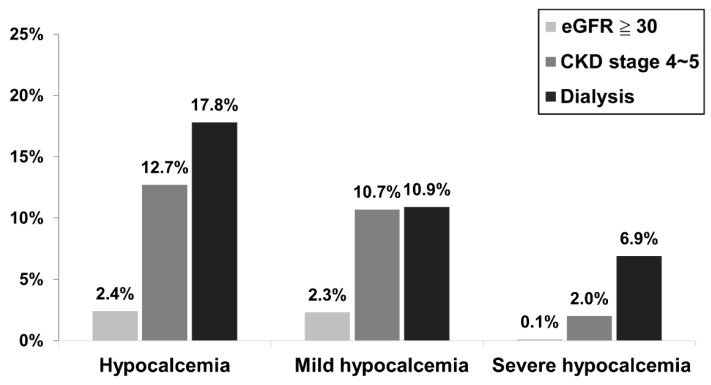
Incidence of hypocalcemia within 30 days post denosumab treatment among different renal function groups on denosumab. Hypocalcemia was defined as a corrected calcium level less than 8.5 mg/dl. Mild hypocalcemia was defined as a corrected calcium level higher than or equal to 7.0 mg/dl and less than 8.5 mg/dl (7.0 ≦ corrected calcium < 8.5 mg/dl). Severe hypocalcemia was defined as a corrected calcium level less than 7.0 mg/dl.

**Figure 2 healthcare-11-00313-f002:**
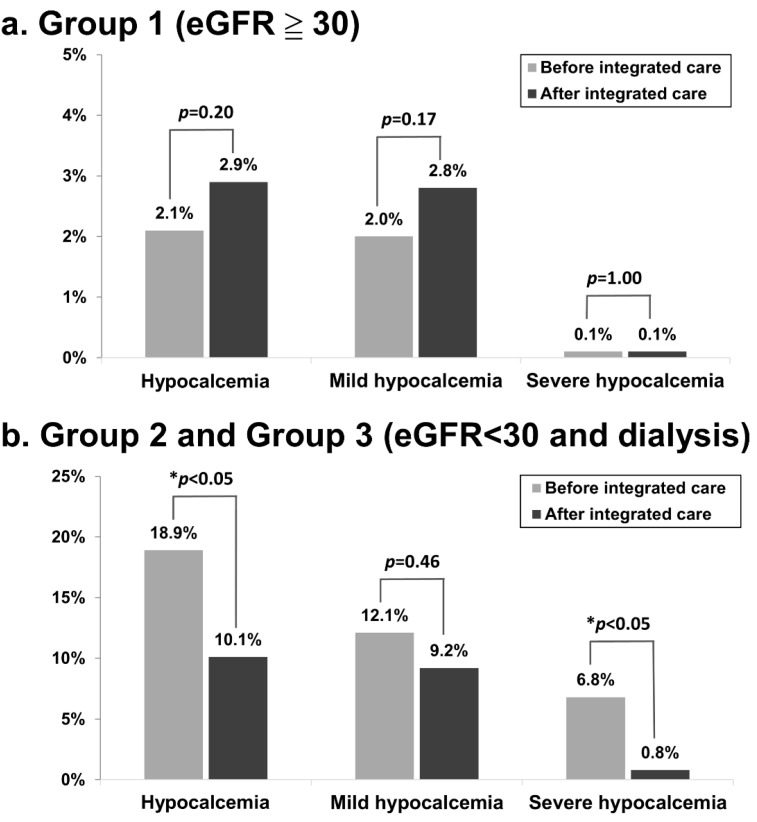
Incidence of hypocalcemia within 30 days post denosumab treatment before and after the integrated care program (**a**) among patients with an eGFR higher than or equal to 30 and (**b**) among patients with advanced CKD and ESRD. Hypocalcemia was defined as a corrected calcium level less than 8.5 mg/dl. Mild hypocalcemia was defined as a corrected calcium level higher than or equal to 7.0 mg/dl and less than 8.5 mg/dl (7.0 ≦ corrected calcium < 8.5 mg/dl). Severe hypocalcemia was defined as a corrected calcium level less than 7.0 mg/dl.

**Figure 3 healthcare-11-00313-f003:**
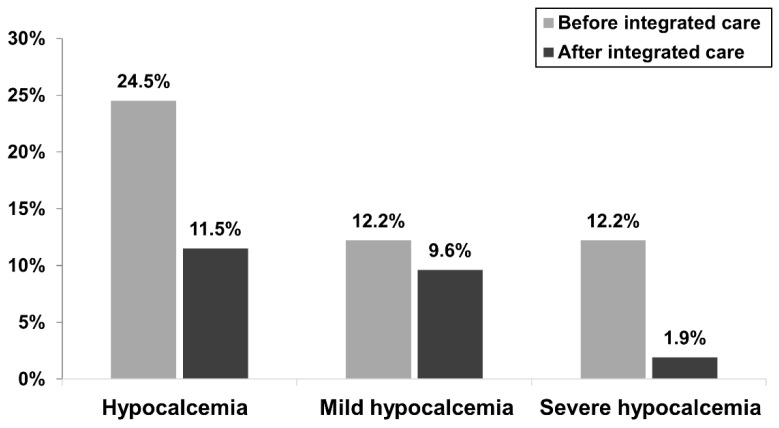
Incidence of post-treatment hypocalcemia within 30 days before and after the integrated care program implementation among dialysis patients. Hypocalcemia was defined as a corrected calcium level less than 8.5 mg/dl. Mild hypocalcemia was defined as a corrected calcium level higher than or equal to 7.0 mg/dl and less than 8.5 mg/dl (7.0 ≦ corrected calcium < 8.5 mg/dl). Severe hypocalcemia was defined as a corrected calcium level less than 7.0 mg/dl.

**Table 1 healthcare-11-00313-t001:** Clinical characteristics of the patients that received their first dose of denosumab.

	Group 1 eGFR ≥ 30 (n = 2957)		Group 2 CKD Stage 4 and 5 (n = 150)		Group 3 Dialysis (n = 101)		*p* Value	H-Statistic/ Chi-Square Statistic
Demographic characteristics								
Age †	73.5	(65.9–80.8)	79.6	(72.2–86.2)	69.0	(61.0–76.1)	<0.001 **	52.003
Male sex ‡	543	(18.4%)	33	(22.0%)	26	(25.7%)	0.102	4.569
Weight †	55.4	(49.7–62.0)	57.8	(47.1–65.0)	54.0	(48.0–63.2)	0.567	1.135
Height †	154.0	(149.4–159.0)	153.0	(148.6–159.4)	156.5	(150.1–161.9)	0.193	3.290
BMI †	23.3	(20.9–26.0)	23.3	(20.2–27.3)	23.0	(20.3–24.8)	0.382	1.923
Comorbidity								
Diabetes ‡	1002	(33.9%)	75	(50.0%)	48	(47.5%)	<0.001 **	23.386
Hypertension ‡	1357	(45.9%)	120	(80.0%)	88	(87.1%)	<0.001 **	127.841
Stroke ‡	423	(14.3%)	35	(23.3%)	22	(21.8%)	0.002 **	12.957
CAD ‡	827	(28.0%)	50	(33.3%)	57	(56.4%)	<0.001 **	39.709
PAD ‡	263	(8.9%)	19	(12.7%)	21	(20.8%)	<0.001 **	18.074
CHF ‡	331	(11.2%)	21	(18.0%)	42	(41.6%)	<0.001 **	87.056
Baseline laboratory data								
eGFR §	78.3	(62.4–94.3)	21.9	(14.1–26.4)	-	-	<0.001 **	428.137
Corrected Calcium †	9.10	(8.82–9.40)	9.00	(8.50–9.36)	9.35	(8.81–10.10)	<0.001 **	18.146
Phosphate †	3.6	(3.3–4.0)	4.0	(3.5–4.5)	4.4	(3.7–5.9)	<0.001 **	64.366
ALP †	103.0	(81.0–134.0)	127.0	(96.3–174.8)	135.5	(106.0–189.0)	<0.001 **	40.994
Intact-PTH †	52.3	(37.6–71.3)	178.0	(84.2–303.9)	341.9	(124.3–677.3)	<0.001 **	135.188
Post-treatment hypocalcemia within 30 days following the first dose of denosumab treatment								
Hypocalcemia ‡	71	(2.4%)	19	(12.7%)	18	(17.8%)	<0.001 **	113.224
Mild hypocalcemia ‡	68	(2.3%)	16	(10.7%)	11	(10.9%)	<0.001 **	57.598
Severe hypocalcemia ‡	3	(0.1%)	3	(2.0%)	7	(6.9%)	<0.001 **	122.774

* *p* < 0.05, ** *p* < 0.01. † Kruskal–Wallis test. ‡ Chi-square test. § Mann–Whitney test. Values are expressed as number (percentage) or median (interquartile range). eGFR, estimated glomerular filtration rate; BMI, body mass index; CAD, coronary artery disease; PAD, peripheral artery disease; CHF, congestive heart failure; ALP, alkaline phosphatase; PTH, parathyroid hormone. Hypocalcemia was defined as a corrected calcium level less than 8.5 mg/dl. Mild hypocalcemia was defined as a corrected calcium level higher than or equal to 7.0 mg/dl and less than 8.5 mg/dl (7.0 ≦ corrected calcium < 8.5 mg/dl). Severe hypocalcemia was defined as a corrected calcium level less than 7.0 mg/dl.

**Table 2 healthcare-11-00313-t002:** Factors associated with severe hypocalcemia among patients within 30 days after receiving first dose of denosumab treatment.

	Univariate			Multivariate		
	Hazard Ratio	95% CI	*p* Value	Hazard Ratio	95% CI	*p* Value
Integrated care	0.26	(0.06–1.19)	0.083			
Renal function group	8.18	(4.43–15.08)	<0.001 **	8.31	(1.20-57.30)	0.032 *
Age	0.94	(0.90–0.98)	0.003 **			
Male sex	1.30	(0.36–4.74)	0.687			
BMI	0.81	(0.67–0.98)	0.027 *			
Diabetes	2.17	(0.73–6.46)	0.163			
Hypertension	5.82	(1.29–26.26)	0.021 *			
Stroke	2.55	(0.79–8.28)	0.119			
CAD	1.10	(0.34–3.57)	0.877			
PAD	2.89	(0.79–10.49)	0.107			
CHF	1.30	(0.29–5.88)	0.732			
Baseline corrected Ca	0.25	(0.13–0.49)	<0.001 **	0.29	(0.09–0.93)	0.038 *
Baseline phosphate	1.99	(1.48–2.67)	<0.001 **			
Baseline ALP	1.01	(1.00–1.01)	<0.001 **			
Baseline intact-PTH	1.00	(1.00–1.00)	<0.001 **	1.00	(1.00–1.00)	0.096

* *p* < 0.05; ** *p* < 0.01. Cox proportional hazard regression. CI confidence interval; BMI, body mass index; CAD, coronary artery disease; PAD, peripheral artery disease; CHF, congestive heart failure; ALP alkaline phosphatase; PTH parathyroid hormone. Severe hypocalcemia was defined as a corrected calcium level less than 7.0 mg/dl.

**Table 3 healthcare-11-00313-t003:** Comparison of the clinical characteristics before and after integrated care among patients with advanced CKD and ESRD (the target group of our integrated care).

	Before Integrated Care (n = 132)		After Integrated Care (n = 119)		*p* Value	Z-Statistic/ Chi-Square Statistic
Demographic characteristics						
Age †	75.2	(65.5–83.8)	76.1	(67.2–84.6)	0.206	−1.264
Male sex ‡	31	(23.5%)	28	(23.5%)	0.993	0.000
Weight †	55.3	(43.9–62.5)	56.2	(48.5–65.0)	0.203	−1.273
Height †	153.0	(147.2–160.3)	155.8	(151.5–159.8)	0.660	−0.440
BMI †	23.0	(19.8–26.5)	23.3	(20.8–25.6)	0.430	−0.788
Renal function group						
CKD stage 4 and 5 ‡	83	(62.9%)	67	(56.3%)	0.290	1.121
Dialysis ‡	49	(37.1%)	52	(47.3%)	0.290	1.121
Comorbidity						
Diabetes ‡	57	(43.2%)	66	(55.5%)	0.052	3.762
Hypertension ‡	107	(81.1%)	101	(84.9%)	0.424	0.638
Stroke ‡	29	(22.0%)	28	(23.5%)	0.769	0.086
CAD ‡	54	(40.9%)	53	(44.5%)	0.562	0.336
PAD ‡	23	(17.4%)	17	(14.3%)	0.498	0.458
CHF ‡	32	(24.2%)	37	(31.1%)	0.226	1.467
Baseline laboratory data						
eGFR (CKD stage 4 and 5) †	22.0	(14.25–27.0)	21.0	(14.0–25.0)	0.704	−0.380
Corrected Ca †	9.0	(8.6–9.4)	9.2	(8.7–9.4)	0.369	−0.898
Phosphate †	4.2	(3.7–4.9)	4.1	(3.5–5.2)	0.964	−0.045
ALP †	129.0	(105.0–175.0)	128.0	(96.8–180.8)	0.977	−0.029
Intact-PTH †	227.2	(83.9–570.2)	246.3	(124.5–629.3)	0.984	−0.001
Post-treatment hypocalcemia within 30 days following the first dose of denosumab treatment						
Hypocalcemia ‡	25	(18.9%)	12	(10.1%)	0.049 *	3.899
Mild hypocalcemia ‡	16	(12.1%)	11	(9.2%)	0.463	0.538
Severe hypocalcemia §	9	(6.8%)	1	(0.8%)	0.021 *	5.823

* *p* < 0.05, ** *p* < 0.01. † Wilcoxon signed-rank test. ‡ Chi-square test. § Fisher’s exact test. Values are expressed as number (percentage) or median (interquartile range). eGFR estimated glomerular filtration rate; BMI, body mass index; CAD, coronary artery disease; PAD, peripheral artery disease; CHF, congestive heart failure; ALP, alkaline phosphatase; PTH parathyroid hormone. Hypocalcemia was defined as a corrected calcium level less than 8.5 mg/dl. Mild hypocalcemia was defined as a corrected calcium level higher than or equal to 7.0 mg/dl and less than 8.5 mg/dl (7.0 ≦ corrected calcium < 8.5 mg/dl). Severe hypocalcemia was defined as a corrected calcium level less than 7.0 mg/dl.

**Table 4 healthcare-11-00313-t004:** Factors associated with severe hypocalcemia within 30 days following the first dose of denosumab treatment among patients with advanced CKD and ESRD (the target group of our integrated care program).

	Univariate			Multivariate		
	Hazard Ratio	95% CI	*p* Value	Hazard Ratio	95% CI	*p* Value
Integrated care	0.12	(0.02–0.95)	0.045 *	0.15	(0.02–1.23)	0.077
Renal function group	3.74	(0.97–14.46)	0.056	2.66	(0.46–15.38)	0.275
Age	0.94	(0.90–0.98)	0.006 **	0.97	(0.91–1.02)	0.222
Male sex	0.82	(0.17–3.86)	0.802	0.99	(0.20–5.03)	0.993
BMI	0.82	(0.67–1.01)	0.057	0.87	(0.70–1.07)	0.177
Diabetes	1.08	(0.31–3.73)	0.903			
Hypertension	1.98	(0.25–15.62)	0.517			
Stroke	2.32	(0.66–8.23)	0.192			
CAD	0.94	(0.27–3.33)	0.924			
PAD	1.33	(0.28–6.28)	0.716			
CHF	0.71	(0.15–3.34)	0.663			
Baseline corrected Ca	0.54	(0.24–0.49)	0.129			
Baseline phosphate	1.35	(0.88–2.06)	0.165			
Baseline ALP	1.00	(1.00–1.01)	0.347			
Baseline intact-PTH	1.00	(1.00–1.00)	0.311			

* *p* < 0.05; ** *p* < 0.01. Cox proportional hazard regression. CI confidence interval; BMI, body mass index; CAD, coronary artery disease; PAD, peripheral artery disease; CHF, congestive heart failure; ALP alkaline phosphatase; PTH parathyroid hormone. Severe hypocalcemia was defined as a corrected calcium level less than 7.0 mg/dl.

## Data Availability

Not applicable.

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
