# Peer review of "Integrated Osteoporosis Care to Reduce Denosumab-Associated Hypocalcemia for Patients with Advanced Chronic Kidney Disease and End-Stage Renal Disease"

_healthcare, 2023, doi:10.3390/healthcare11030313_

Round 1
Reviewer 1 Report
In this study, the authors conducted a retrospective study to compare the incidence rate of denosumab-associated hypocalcemia before and after the integrated care program with medical records with patients on their first dose of denosumab treatment from January 2012 to December 2021, including 101 dialysis patients, 150 patients with advanced CKD (stage 4 and 5), and 2,957 patients with eGFR more than or equal to 30, respectively. After analysis, the authors concluded that multidisciplinary integrated care with approaches to reduce the hypocalcemia complication of denosumab use in patients with advanced CKD, and ESRD could improve patient safety.
It seems that this study holds some clinical significance. Yet, lots of concerns remain to be addressed.
Major issues:
1. The study significance is not clear enough.
2. The authors mentioned that the aim of this study is to compare the incidence rate of denosumab-associated hypocalcemia before and after the integrated care program. Do the authors want to make within-group comparisons?
3. How did the authors calculate the sample sizes?
4. How did the authors define “integrated care program”?
5. Since it’s a retrospective study, I think the tones in the Conclusion should be scaled down.
6. I think the statistical method is a way to simple and easy. I suggest the authors consider regression models.
Minor issue:
1. The authors should take more updated references.
2. In Table 1, the authors should show us the statistics, such as F-value, c2 value.
3. The Inclusion criteria and Exclusion criteria are not strict or clear enough, and what about the quality control?
4. Did the authors exclude the interference of confounding factors?
Author Response
Response to the comments
Dear reviewer 1: 
Thank you for your detailed review. We feel that your insightful suggestions helped us in improving the manuscript. We have provided a point-by-point response to all your comments below. We revise our manuscript using a word processing program. In the revised manuscript, all the changes are highlighted using track changes to make them more visible. Your original comments are listed below followed by our response to each comment.
Reviewer 1
In this study, the authors conducted a retrospective study to compare the incidence rate of denosumab-associated hypocalcemia before and after the integrated care program with medical records with patients on their first dose of denosumab treatment from January 2012 to December 2021, including 101 dialysis patients, 150 patients with advanced CKD (stage 4 and 5), and 2,957 patients with eGFR more than or equal to 30, respectively. After analysis, the authors concluded that multidisciplinary integrated care with approaches to reduce the hypocalcemia complication of denosumab use in patients with advanced CKD, and ESRD could improve patient safety.
It seems that this study holds some clinical significance. Yet, lots of concerns remain to be addressed.
Major issues:
Comment:
- The study significance is not clear enough.
Reply:
Yes. We deeply appreciate your valuable opinion. We apologize that we did not clearly explain the design of our study. The incidence of hypocalcemia is higher in patients with advanced CKD and ESRD on denosumab treatment, so our integrated care program focused on patients with advanced CKD and ESRD. We analyzed the incidence of post-treatment hypocalcemia within 30 days following the first dose of denosumab treatment among different renal function groups to confirm the target patient group of integrated care program. Mild hypocalcemia may be asymptomatic or cause muscle cramps, but severe hypocalcemia could be life-threatening. Severe hypocalcemia is highly associated with patient safety, so we focused on the incidence rate of denosumab-associated severe hypocalcemia in this study. Then, we analyzed the incidence of severe hypocalcemia within 30 days following the first dose of denosumab treatment before and after the implementation of integrated care program, especially in the subgroup of advanced CKD (eGFR less than 30) and ESRD. Previous study by Chen et al. (The Journal of Clinical Endocrinology & Metabolism 2014, 99, 2426-2432) suggested some approaches to reduce the hypocalcemia complication, including (1) calcium and vitamin D supplement during denosumab treatment; and (2) increase the dialysate calcium concentration during denosumab treatment for dialysis patients. We also implemented these approaches in our integrated care program. To our knowledge, previous studies seldom compared the incidence of severe hypocalcemia after implementation of these approaches. Our result showed the incidence of post-treatment severe hypocalcemia within 30 days in advanced CKD and dialysis patients was significantly lower after the integrated care program (6.8% vs. 0.8%, p < 0.05). Not surprisingly, our integrated care program did not reduce the incidence of hypocalcemia within 30 days following the first dose of denosumab treatment among patients with eGFR more than or equal to 30.
Our study showed patients on denosumab with poor renal function were younger owing to the CKD-MBD related secondary osteoporosis. In the univariate regression of Cox proportional hazards model, patients with younger age were associated with severe hypocalcemia. This may be related to these younger patients were mainly dialysis patients. The poor renal function was remained in the multivariate regression model. The poor renal function is a stronger factor to the development of severe hypocalcemia in our Table 2. However, Table 4 revealed poor renal function was not a significant risk factor for severe hypocalcemia. This finding is similar to the study reported by Saito et al. They reported that renal impairment is not a risk factor for denosumab-induced hypocalcemia in a strict denosumab administration management system with calcium and vitamin D supplementation.
(Please see the method and discussion sections in our clean copy of revised manuscript)
- The authors mentioned that the aim of this study is to compare the incidence rate of denosumab-associated hypocalcemia before and after the integrated care program. Do the authors want to make within-group comparisons?
Reply:
Yes. We deeply appreciate your valuable opinion. We apologize that we did not clearly explain the design of our study. To our knowledge, few studies compared the incidence of severe hypocalcemia after implementation of these approaches. Our study mainly focused on the severe hypocalcemia within 30 days following the first dose of denosumab treatment before and after the implementation of integrated care program among patients with advanced CKD and ESRD. We want to make comparisons between the severe hypocalcemia before integrated care program and the severe hypocalcemia after integrated care program.
(Please see the method section in our clean copy of revised manuscript)
- How did the authors calculate the sample sizes?
Reply:
Yes. Thank you for your comment. We apologize that we did not clearly explain the design of our study. We retrospectively analyzed 10 years of medical records in patients on their first dose of denosumab treatment from January 2012 to December 2021 in our hospital. Our study mainly focused on the incidence of severe hypocalcemia within 30 days following the first dose of denosumab treatment before and after the implementation of integrated care program, especially in the subgroup of advanced CKD and ESRD. We did not do the random sampling. A total of 15,413 patients in our osteoporosis database between January 2012 to December 2021. Based on the baseline renal function, there were 1,078 patients with advanced CKD and ESRD; and 14,335 patients with eGFR more than or equal to 30 ml/min/1.73m2. According to previous studies, the incidence of hypocalcemia after denosumab treatment in ESRD patients ranges from 25.5% to 63%, but no study reported the incidence of hypocalcemia after denosumab treatment in patients with advanced CKD (eGFR <30). However, the incidence of hypocalcemia after denosumab treatment in patients with advanced CKD (eGFR <30) would smaller than in patients with ERSD. We assumed that the incidence of hypocalcemia after denosumab treatment in patients with advanced CKD and ESRD is around 30% in our hospital (so the population proportion is 0.3). We also assumed the confidence level is 95%, and a margin of error is 5%. For 1,078 patients with advanced CKD and ESRD, 249 or more patients are needed to have a confidence level of 95% that the real value is within ± 5% of the measured/surveyed value. There are 251 patients with advanced CKD and ESRD in our study.
- How did the authors define “integrated care program”?
Reply:
Yes. Thank you for your advice. We apologize that we did not clearly explain the integrated osteoporosis care program in our study. As suggested, we add a subsection "2.1. Integrated osteoporosis care program among patients with advanced CKD and ESRD" in the section methods.
The incidence of hypocalcemia is higher in patients with advanced CKD and ESRD on denosumab treatment, so our integrated care program focused on patients with advanced CKD and ESRD. We had been built up a multidisciplinary integrated care pro-gram for osteoporosis among patients with advanced CKD and ESRD since 2018. We ap-plied the computerized physician order entry (CPOE) system in our electronic health in-formation system (EHIS) and set up a CPOE module for osteoporosis integrated care pro-gram among patients with advanced CKD and ESRD. This module comprised the orders of laboratory examinations, dual-energy X-ray absorptiometry (DEXA) examination, and referral sheet to osteoporosis-qualified nursing consultants (case managers). The Laboratory examinations included renal function, calcium, phosphorus, albumin, alkaline phosphatase, intact parathyroid hormone, and other endocrine profiles related to secondary osteoporosis. Osteoporosis-qualified nursing consultants (case managers) further referred patients with advanced CKD and ESRD to nephrologists for the evaluation of CKD-MBD and patient selection before denosumab treatment.
Besides, we also set up a warning system in our EHIS to ensure the patients on denosumab treatment had been (1) checked baseline calcium and correct hypocalcemia prior to denosumab use; (2) received calcium (3g/day) and vitamin D (0.5 to 2 ug/day) supplement during denosumab treatment; (3) monitored calcium levels after denosumab treatment. All the dialysis patients in our hospital received calcium and vitamin D supplements during denosumab treatment. Moreover, we also prophylactically increased the dialysate calcium concentration during denosumab treatment for both hemodialysis and peritoneal dialysis patients. After that, we checked the calcium and phosphate levels every week in the hemodialysis patients for one month, and every two weeks in the peritoneal dialysis patients for one month.
(Please see Pages 2, lines 88 in our clean copy of revised manuscript)
- Since it’s a retrospective study, I think the tones in the Conclusion should be scaled down.
Reply:
Yes. Thank you for your advice. As suggested, we scaled down our conclusion. Our study showed multidisciplinary integrated care may reduce the incidence rate of denosumab-associated severe hypocalcemia among patients with advanced CKD and ESRD.
(Please see Pages 11, lines 341-349 in our clean copy of revised manuscript)
- I think the statistical method is a way to simple and easy. I suggest the authors consider regression models.
Reply:
Yes. Thank you for your advice. As suggested, we used the Cox proportional hazards regression model to analyze factors that may be associated with severe hypocalcemia within 30 days following the first dose of denosumab treatment.
(Please see the method section in our clean copy of revised manuscript)
Minor issue:
Comment:
- The authors should take more updated references.
Reply:
Yes. Thank you for your advice. As suggested, we take more updated references.
(Please see the reference section in our clean copy of revised manuscript)
- In Table 1, the authors should show us the statistics, such as F-value, c2 value.
Reply:
Yes. Thank you for your advice. As suggested, we showed the H- statistics and Chi-Square statistic in our Table 1. The Kruskal Wallis test is the non-parametric alternative to the One Way ANOVA. The F-statistic results from performing ANOVA. Kruskal-Wallis reports an H-statistic (interpretable in much the same way that the F-statistic is interpreted).
(Please see Table 1 in our clean copy of revised manuscript)
- The Inclusion criteria and Exclusion criteria are not strict or clear enough, and what about the quality control?
Reply:
Yes. We deeply appreciate your valuable opinion. The incidence of hypocalcemia is higher in patients with advanced CKD and ESRD on denosumab treatment, so our integrated care program focused on patients with advanced CKD and ESRD (the target group of our integrated care). As suggested, we used the Cox proportional hazards regression model to analyze factors that may be associated with severe hypocalcemia within 30 days following the first dose of denosumab treatment. In the revised Table 3, the demographic characteristics, percentage of the renal function subgroup, percentage of comorbidities, and baseline laboratory data after the integrated care program were not statistically different compared to before. However, the incidence of hypocalcemia within 30 days post-treatment in advanced CKD and dialysis patients was significantly lower after the integrated care program. However, the retrospective nature of the study may have led to some unrecognized confounding factors to bias the findings. This is a limitation in our study.
(Please see the method and result sections in our clean copy of revised manuscript)
- Did the authors exclude the interference of confounding factors?
Reply:
Yes. Thank you for your advice. We deeply appreciate your valuable opinion. The incidence of hypocalcemia is higher in patients with advanced CKD and ESRD on denosumab treatment, so our integrated care program focused on patients with advanced CKD and ESRD (the target group of our integrated care). In the revised Table 3, the demographic characteristics, percentage of the renal function subgroup, percentage of comorbidities, and baseline laboratory data after the integrated care program were not statistically different compared to before. However, the incidence of hypocalcemia within 30 days post-treatment in advanced CKD and dialysis patients was significantly lower after the integrated care program. However, the retrospective nature of the study may have led to some unrecognized confounding factors to bias the findings. This is a limitation in our study.
(Please see the method and result sections in our clean copy of revised manuscript)

Reviewer 2 Report
This retrospective research manuscript looked at the incidence of hypercalcemia in patients with chronic kidney disease or end-stage renal disease within 30 days of denosumab treatment and following a multidisciplinary integrated care program over a 10-year period at one hospital.
The benefit of this research is unclear to the scientific community. The study population has a selection bias to one location and no effort is made to establish the representative nature of this sample population to a larger demographic. The integrated care program is also poorly described and from what is described, it is unique to the study site. Consequently the study is not relevant outside of the hospital collecting the data. Both issues need to be addressed prior to publication.
For a scientific study to be published there needs to be enough information for the methods to be reproducible. The methods should be expanded to clearly outline what the program includes and how it differed and/or remained the same for the patients included in the study.
One of the few component of the integrated care program that is mentioned (although not in the methods) is that each participant was given supplemental calcium and vitamin D. Details including the doses of each compound given should be added to the methods.
It is possible that all of the results the authors describe are the result of supplemental calcium and vitamin D alone and not any other component of the integrated care program. No aspect of the integrated care program was controlled, tested or evaluated in the context of the literature on the subject.
No control group existed in the study. Nor was data prior to the onset of the integrated care program included in the study. Either is needed to interpret the data.
The authors adequately explain the statistics used in evaluating unpaired data however it is unclear what statistics were used in evaluating paired data (before and after the integrated care program). Please clarify the statistics used.
The authors use an interval or “within 30 days” to define their test period. That can vary from 1-29 days. The number of days is another potential confounding variable that should either be controlled for or used in the analysis to determine if timing in that 30-day period as an effect.
The first paragraph of the discussion does not discuss the results, or data in the study. This background information should be included in the introduction and omitted where redundant. The discussion should evaluate the results in the context of the literature and our understanding of the science involved. What evidence is there on the success of integrated care programs. What aspects of the program may/ or may not have contributed to the results (etcetera).
The section of the conclusions (line 204-207) that addresses healthcare professional awareness should be omitted. This was not the focus of this paper nor was it discussed to the extent that conclusions could or should be drawn in this paper.
The use of abbreviation in journal titles as well as the use of capital letters (or lack thereof) in the journal names and article titles is not consistent in the references. Errors sometimes exist when using citation software. Please review and revise the references section for consistency in meeting journal standards.
Author Response
Dear reviewer 2: 
Thank you for your detailed review. We feel that your insightful suggestions helped us in improving the manuscript. We have provided a point-by-point response to all your comments below. We revise our manuscript using a word processing program. In the revised manuscript, all the changes are highlighted using track changes to make them more visible. Your original comments are listed below followed by our response to each comment.
Reviewer 2
This retrospective research manuscript looked at the incidence of hypercalcemia in patients with chronic kidney disease or end-stage renal disease within 30 days of denosumab treatment and following a multidisciplinary integrated care program over a 10-year period at one hospital.
Comment:
The benefit of this research is unclear to the scientific community. The study population has a selection bias to one location and no effort is made to establish the representative nature of this sample population to a larger demographic. The integrated care program is also poorly described and from what is described, it is unique to the study site. Consequently the study is not relevant outside of the hospital collecting the data. Both issues need to be addressed prior to publication.
Reply:
Yes. We deeply appreciate your valuable opinion. We apologize that we did not clearly explain the design of our study. The incidence of hypocalcemia is higher in patients with advanced CKD and ESRD on denosumab treatment, so our integrated care program focused on patients with advanced CKD and ESRD. We analyzed the incidence of post-treatment hypocalcemia within 30 days following the first dose of denosumab treatment among different renal function groups to confirm the target patient group of integrated care program. Mild hypocalcemia may be asymptomatic or cause muscle cramps, but severe hypocalcemia could be life-threatening. Severe hypocalcemia is highly associated with patient safety, so we focused on the incidence rate of denosumab-associated severe hypocalcemia in this study. Then, we analyzed the incidence of severe hypocalcemia within 30 days following the first dose of denosumab treatment before and after the implementation of integrated care program, especially in the subgroup of advanced CKD (eGFR less than 30) and ESRD. Previous study by Chen et al. (The Journal of Clinical Endocrinology & Metabolism 2014, 99, 2426-2432) suggested some approaches to reduce the hypocalcemia complication, including (1) calcium and vitamin D supplement during denosumab treatment; and (2) increase the dialysate calcium concentration during denosumab treatment for dialysis patients. We also implemented these approaches in our integrated care program. To our knowledge, previous studies seldom compared the incidence of severe hypocalcemia after implementation of these approaches. Our result showed the incidence of post-treatment severe hypocalcemia within 30 days in advanced CKD and dialysis patients was significantly lower after the integrated care program (6.8% vs. 0.8%, p < 0.05). Not surprisingly, our integrated care program did not reduce the incidence of hypocalcemia within 30 days following the first dose of denosumab treatment among patients with eGFR more than or equal to 30.
Our study showed patients on denosumab with poor renal function were younger owing to the CKD-MBD related secondary osteoporosis. In the univariate regression of Cox proportional hazards model, patients with younger age were associated with severe hypocalcemia. This may be related to these younger patients were mainly dialysis patients. The poor renal function was remained in the multivariate regression model. The poor renal function is a stronger factor to the development of severe hypocalcemia in our Table 2. However, Table 4 revealed poor renal function was not a significant risk factor for severe hypocalcemia. This finding is similar to the study reported by Saito et al. They reported that renal impairment is not a risk factor for denosumab-induced hypocalcemia in a strict denosumab administration management system with calcium and vitamin D supplementation.
(Please see the method and discussion sections in our clean copy of revised manuscript)
Comment:
For a scientific study to be published there needs to be enough information for the methods to be reproducible. The methods should be expanded to clearly outline what the program includes and how it differed and/or remained the same for the patients included in the study.
Reply:
Yes. Thank you for your comment. We apologize that we did not clearly explain the integrated osteoporosis care program in our study. As suggested, we add a subsection "2.1. Integrated osteoporosis care program among patients with advanced CKD and ESRD" in the section methods.
The incidence of hypocalcemia is higher in patients with advanced CKD and ESRD on denosumab treatment, so our integrated care program focused on patients with advanced CKD and ESRD. We had been built up a multidisciplinary integrated care pro-gram for osteoporosis among patients with advanced CKD and ESRD since 2018. We ap-plied the computerized physician order entry (CPOE) system in our electronic health in-formation system (EHIS) and set up a CPOE module for osteoporosis integrated care pro-gram among patients with advanced CKD and ESRD. This module comprised the orders of laboratory examinations, dual-energy X-ray absorptiometry (DEXA) examination, and referral sheet to osteoporosis-qualified nursing consultants (case managers). The Laboratory examinations included renal function, calcium, phosphorus, albumin, alkaline phosphatase, intact parathyroid hormone, and other endocrine profiles related to secondary osteoporosis. Osteoporosis-qualified nursing consultants (case managers) further referred patients with advanced CKD and ESRD to nephrologists for the evaluation of CKD-MBD and patient selection before denosumab treatment.
Besides, we also set up a warning system in our EHIS to ensure the patients on denosumab treatment had been (1) checked baseline calcium and correct hypocalcemia prior to denosumab use; (2) received calcium (3g/day) and vitamin D (0.5 to 2 ug/day) supplement during denosumab treatment; (3) monitored calcium levels after denosumab treatment. All the dialysis patients in our hospital received calcium and vitamin D supplements during denosumab treatment. Moreover, we also prophylactically increased the dialysate calcium concentration during denosumab treatment for both hemodialysis and peritoneal dialysis patients. After that, we checked the calcium and phosphate levels every week in the hemodialysis patients for one month, and every two weeks in the peritoneal dialysis patients for one month.
(Please see method section in our clean copy of revised manuscript)
Comment:
One of the few components of the integrated care program that is mentioned (although not in the methods) is that each participant was given supplemental calcium and vitamin D. Details including the doses of each compound given should be added to the methods.
Reply:
Yes. Thank you for your advice. We added the doses of each compound to the methods. (2) received calcium (3g/day) and vitamin D (0.5 to 2 ug/day) supplement during denosumab treatment.
(Please see Pages 3, lines 105-106 in our clean copy of revised manuscript)
Comment:
It is possible that all of the results the authors describe are the result of supplemental calcium and vitamin D alone and not any other component of the integrated care program. No aspect of the integrated care program was controlled, tested or evaluated in the context of the literature on the subject.
Reply:
Yes. Thank you for your advice. We apologize that we did not clearly explain the integrated osteoporosis care program in our study. As suggested, we add a subsection "2.1. Integrated osteoporosis care program among patients with advanced CKD and ESRD" in the section methods.
(Please see method section in our clean copy of revised manuscript)
Comment:
No control group existed in the study. Nor was data prior to the onset of the integrated care program included in the study. Either is needed to interpret the data.
Reply:
Yes. We deeply appreciate your valuable opinion. We apologize that we did not clearly explain the design of our study. We had been built up a multidisciplinary integrated care program for osteoporosis among patients with advanced CKD and ESRD since 2018. The study cohort was divided into the following two parts: (1) the data from January 2012 to December 2017 as the data set of "before integrated care"; and (2) the data from January 2018 to December 2021 as the set of "after integrated care". According to previous studies, the incidence of hypocalcemia is higher in patients with advanced CKD and ESRD on denosumab treatment. We analyzed the incidence of post-treatment hypocalcemia within 30 days following the first dose of denosumab treatment among different renal function groups to confirm the target patient group of integrated care program. Then, we analyzed the incidence of severe hypocalcemia within 30 days following the first dose of denosumab treatment before and after the implementation of integrated care program, especially in the subgroup of advanced CKD (eGFR less than 30) and ESRD.
(Please see method section in our clean copy of revised manuscript)
Comment:
The authors adequately explain the statistics used in evaluating unpaired data however it is unclear what statistics were used in evaluating paired data (before and after the integrated care program). Please clarify the statistics used.
Reply:
Yes. Thank you for your advice. As suggested, we revised the method in evaluating paired data (before and after the integrated care program). We used the Wilcoxon signed-rank test for non-parametric continuous variables of paired data.
(Please see method section in our clean copy of revised manuscript)
Comment:
The authors use an interval or “within 30 days” to define their test period. That can vary from 1-29 days. The number of days is another potential confounding variable that should either be controlled for or used in the analysis to determine if timing in that 30-day period as an effect.
Reply:
Yes. Thank you for your comment. Because the half-life of denosumab is around 30 days and clinical hypocalcemia tends to occur within the 7–20 days following the first dose of denosumab, we monitored the hypocalcemia within 30 days following the first dose of denosumab as our outcome (calcium level monitoring as protocol shown in 2.1. Integrated osteoporosis care program among patients with advanced CKD and ESRD).
(Please see method section in our clean copy of revised manuscript)
Comment:
The first paragraph of the discussion does not discuss the results, or data in the study. This background information should be included in the introduction and omitted where redundant. The discussion should evaluate the results in the context of the literature and our understanding of the science involved. What evidence is there on the success of integrated care programs. What aspects of the program may/ or may not have contributed to the results (etcetera).
Reply:
Yes. We deeply appreciate your valuable opinion. Thank you for your advice. As suggested, we add more discussion to evaluate the results.
Our study showed patients on denosumab with poor renal function were younger owing to the CKD-MBD related secondary osteoporosis. In the univariate regression of Cox proportional hazards model, patients with younger age were associated with severe hypocalcemia. This may be related to these younger patients were mainly dialysis patients. The poor renal function was remained in the multivariate regression model. The poor renal function is a stronger factor to the development of severe hypocalcemia in our Table 2. However, Table 4 revealed poor renal function was not a significant risk factor for severe hypocalcemia. This finding is similar to the study reported by Saito et al. They reported that renal impairment is not a risk factor for denosumab-induced hypocalcemia in a strict denosumab administration management system with calcium and vitamin D supplementation.
(Please see discussion section in our clean copy of revised manuscript)
Comment:
The section of the conclusions (line 204-207) that addresses healthcare professional awareness should be omitted. This was not the focus of this paper nor was it discussed to the extent that conclusions could or should be drawn in this paper.
Reply:
Yes. We deeply appreciate your valuable opinion. As suggested, we have added more descriptions about the awareness of denosumab-associated hypocalcemia in the discussion section.
Our integrated care program also included a warning system to ensure the patients on denosumab treatment had been (1) checked baseline calcium and correct hypocalcemia prior to denosumab use; (2) received calcium (3g/day) and vitamin D (0.5 to 2 ug/day) supplement during denosumab treatment; (3) monitored calcium levels after denosumab treatment. The warning system may increase the Doctors’ and healthcare professionals’ awareness of denosumab-associated hypocalcemia.
(Please see discussion section in our clean copy of revised manuscript)
Comment:
The use of abbreviation in journal titles as well as the use of capital letters (or lack thereof) in the journal names and article titles is not consistent in the references. Errors sometimes exist when using citation software. Please review and revise the references section for consistency in meeting journal standards.
Reply:
Yes. Thank you for your advice. As suggested, we had revised the references section.
(Please see reference section in our clean copy of revised manuscript)

Reviewer 3 Report
The present submission revealed that the integrated osteoporosis care may reduce Denosumab-induced hypocalcemia in patients with CKD-ESRD. A few concerns about the current version.
1) In the abstract, the phrase 'According to previous studies' should be eliminated or rewritten.
2) The authors should provide that what measures have been taken for the integrated osteoporosis care program in the section methods.
3) In the figures, what is the meaning of 'any hypocalcemia'?
Author Response
Dear reviewer 3: 
Thank you for your detailed review. We feel that your insightful suggestions helped us in improving the manuscript. We have provided a point-by-point response to all your comments below. We revise our manuscript using a word processing program. In the revised manuscript, all the changes are highlighted using track changes to make them more visible. Your original comments are listed below followed by our response to each comment.
Reviewer 3
The present submission revealed that the integrated osteoporosis care may reduce Denosumab-induced hypocalcemia in patients with CKD-ESRD. A few concerns about the current version.
Comment:
- In the abstract, the phrase 'According to previous studies' should be eliminated or rewritten.
Reply:
Yes. Thank you for your advice. As suggested, we removed the phrase "According to previous studies" in the abstract.
(Please see Pages 1, lines 13 in our clean copy of revised manuscript)
Comment:
- The authors should provide that what measures have been taken for the integrated osteoporosis care program in the section methods.
Reply:
Yes. Thank you for your advice. We apologize that we did not clearly explain the integrated osteoporosis care program in our study. As suggested, we add a subsection "2.1. Integrated osteoporosis care program among patients with advanced CKD and ESRD" in the section methods.
The incidence of hypocalcemia is higher in patients with advanced CKD and ESRD on denosumab treatment, so our integrated care program focused on patients with advanced CKD and ESRD. We had been built up a multidisciplinary integrated care pro-gram for osteoporosis among patients with advanced CKD and ESRD since 2018. We ap-plied the computerized physician order entry (CPOE) system in our electronic health in-formation system (EHIS) and set up a CPOE module for osteoporosis integrated care pro-gram among patients with advanced CKD and ESRD. This module comprised the orders of laboratory examinations, dual-energy X-ray absorptiometry (DEXA) examination, and referral sheet to osteoporosis-qualified nursing consultants (case managers). The Laboratory examinations included renal function, calcium, phosphorus, albumin, alkaline phosphatase, intact parathyroid hormone, and other endocrine profiles related to secondary osteoporosis. Osteoporosis-qualified nursing consultants (case managers) further referred patients with advanced CKD and ESRD to nephrologists for the evaluation of CKD-MBD and patient selection before denosumab treatment.
Besides, we also set up a warning system in our EHIS to ensure the patients on denosumab treatment had been (1) checked baseline calcium and correct hypocalcemia prior to denosumab use; (2) received calcium (3g/day) and vitamin D (0.5 to 2 ug/day) supplement during denosumab treatment; (3) monitored calcium levels after denosumab treatment. All the dialysis patients in our hospital received calcium and vitamin D supplements during denosumab treatment. Moreover, we also prophylactically increased the dialysate calcium concentration during denosumab treatment for both hemodialysis and peritoneal dialysis patients. After that, we checked the calcium and phosphate levels every week in the hemodialysis patients for one month, and every two weeks in the peritoneal dialysis patients for one month.
(Please see method section in our clean copy of revised manuscript)
Comment:
- In the figures, what is the meaning of 'any hypocalcemia'?
Reply:
Yes. Thank you for your comment. As suggested, we revised the title of category from "any hypotension" to "hypocalcemia. Besides, we also add more descriptions of category definition in our figure legends. In this study, hypocalcemia was defined as corrected calcium less than 8.5 mg/dl. Mild hypocalcemia was defined as corrected calcium more than or equal to 7.0 mg/dl, and less than 8.5 mg/dl (7.0≦corrected calcium < 8.5 mg/dl). Severe hypocalcemia was defined as corrected calcium less than 7.0mg/dl.
(Please see figures and figure legends)

Round 2
Reviewer 1 Report
After revision, I think the authors have addressed many of my concerns. Yet, some issues remain to be handled.
1. In Table 3, the authors should show us the statistics.
2. Is it appropriate to apply Cox proportional hazards regression model since it’s a retrospective study? The authors should explain it.
3. I think the authors should mention factors associated with severe hypocalcemia within 30 days in the Abstract.
Author Response
Response to the comments
Dear reviewer 1: 
Thank you for your detailed review again. We feel that your insightful suggestions helped us in improving the manuscript. We have provided a point-by-point response to all your comments below. We revise our manuscript using a word processing program. In the revised manuscript, all the changes are highlighted using track changes to make them more visible. Your original comments are listed below followed by our response to each comment.
Reviewer 1
Comment:
- In Table 3, the authors should show us the statistics.
Reply:
Yes. Thank you for your advice. As suggested, we showed the Z-statistics and Chi-Square statistic in our Table 3.
(Please see Table 3 in our clean copy of revised manuscript)
Comment:
- Is it appropriate to apply Cox proportional hazards regression model since it’s a retrospective study? The authors should explain it.
Reply:
Yes. We deeply appreciate your valuable opinion. The Cox regression analysis is a fundamental statistical method for addressing etiological and prognostic hypotheses. It is based on estimating the HR associated with a specific risk factor or predictor for a given endpoint. In survival analysis, both Kaplan–Meier analysis and Cox regression methods are used to address etiological and prognostic hypotheses in clinical and epidemiological research (please see Reference 1 below). There are some literatures using this model to address etiological and prognostic hypotheses in retrospective studies, such as (1). Moriguchi et al. (please see Reference 2 below) compared the logistic regression and the Cox proportional hazard models applied to patients who underwent curative gastrectomy. Although regression coefficients are not all the same, the same risk factors proved significant in both multivariate analyses. This equation for risk factors for prognosis is approached when searching for an appropriate method of retrospective studies using multivariate analyses. (2). Bramante et al. (please see Reference 3 below) also applied Cox proportional hazard models to Metformin and risk of mortality in patients hospitalised with COVID-19 in a retrospective cohort.
The Cox regression analysis not only used in survival analysis, but also be applied to address etiological and prognostic hypotheses (with a specific risk factor or predictor for a given endpoint). Kunizawa et al. (please see Reference 4 below) used the Cox proportional hazard model to evaluate the risk factors for the maximum calcium decline from baseline after denosumab injection in dialysis patients.
Reference:
- Samar Abd ElHafeez, Graziella D’Arrigo, Daniela Leonardis, Maria Fusaro, Giovanni Tripepi, Stefanos Roumeliotis, "Methods to Analyze Time-to-Event Data: The Cox Regression Analysis", Oxidative Medicine and Cellular Longevity, vol. 2021, Article ID 1302811, 6 pages, 2021. https://doi.org/10.1155/2021/1302811
- Moriguchi S, Hayashi Y, Nose Y, Maehara Y, Korenaga D, Sugimachi K. A comparison of the logistic regression and the Cox proportional hazard models in retrospective studies on the prognosis of patients with gastric cancer. J Surg Oncol. 1993 Jan;52(1):9-13. doi: 10.1002/jso.2930520104. PMID: 8441266.
- Bramante, C. T., Ingraham, N. E., Murray, T. A., Marmor, S., Hovertsen, S., Gronski, J., ... & Tignanelli, C. J. (2021). Metformin and risk of mortality in patients hospitalised with COVID-19: a retrospective cohort analysis. The Lancet Healthy Longevity, 2(1), e34-e41.
- Kunizawa, K., Hiramatsu, R., Hoshino, J. et al. Denosumab for dialysis patients with osteoporosis: A cohort study. Sci Rep 10, 2496 (2020). https://doi.org/10.1038/s41598-020-59143-8
Comment:
- I think the authors should mention factors associated with severe hypocalcemia within 30 days in the Abstract.
Reply:
Yes. Thank you for your advice. As suggested, we mention factors associated with severe hypocalcemia within 30 days in the Abstract section. Based on the results of the multivariate regression model, poor renal function (p < 0.05) and lower baseline corrected calcium level (p < 0.05) were associated with severe hypocalcemia within 30 days following the first dose of denosumab treatment.
(Please see the Abstract section in our clean copy of revised manuscript)

Reviewer 2 Report
Thank you for the revisions that were made.
The 1st comments that I made addressed the selection bias at this one location. The authors did not respond or address this comment. But rather recapped the study design and findings.
No control group exists in the study. This comment was made last time. The author’s response was to repeat the study design recap given to my 1st comment.
The “integrated care program” appears to include (if not consistent primarily of) a calcium monitoring and supplementation program. It would appear that aspect of the program was at least consistent and could be evaluated against a control group. The aspects of the program “including consultations” are poorly explained, not standardized and their clinical relevance unknown.
The average day (with standard error) of hypocalcemia testing should be included. As the authors noted day 6 and 29 are within that 30 day window but may not be when hypocalcemia tends to occur. Nor is it clear that the level of hypocalcemia anticipated in constant during that period of time. Results of this study may be due to timing variation between groups.
Sections that were not relevant to the study design were suggested to be omitted. The authors elected to add more on these unrelated topics which is counter to the recommendation.
As requested, the discussion should evaluate the results in the context of the literature, specifically what evidence is there on the success of integrated care programs. What aspects of the program may/ or may not have contributed to the results (etc).
The study has significant number of limitations and biases that must be delineated (many discussed by my fellow reviewers and myself). I would consider accepting if these limitations are appropriately added to the discussion.
Revisions are in need of English spelling and grammar review
Author Response
Dear reviewer 2: 
Thank you for your detailed review again. We feel that your insightful suggestions helped us in improving the manuscript. We have provided a point-by-point response to all your comments below. We revise our manuscript using a word processing program. In the revised manuscript, all the changes are highlighted using track changes to make them more visible. Your original comments are listed below followed by our response to each comment.
Reviewer 2
Comment:
The 1st comments that I made addressed the selection bias at this one location. The authors did not respond or address this comment. But rather recapped the study design and findings.
Reply:
Yes. Thank you for your comment. We deeply appreciate your valuable opinion. We apologize that we did not respond appropriately on last major revision. As suggested, we mentioned this selection bias in the limitation paragraph of discussion section. The study population has a selection bias for one location. The case number of advanced CKD and ESRD in this study is relatively small because this study is a single-center retrospective study. Further studies with a larger sample size are required to confirm the efficacy of these approaches for reducing hypocalcemia complications.. The retrospective nature of the study may have led to some unrecognized confounding factors to bias the findings.
(Please see the limitation paragraph of discussion section in our clean copy of revised manuscript)
Comment:
No control group exists in the study. This comment was made last time. The author’s response was to repeat the study design recap given to my 1st comment.
Reply:
Yes. Thank you for your comment. We deeply appreciate your valuable opinion. We apologize that we did not respond appropriately on last major revision. The aim of this study is to compare the incidence rate of denosumab-associated severe hypocalcemia among patients with advanced CKD and ESRD before and after the implementation of integrated care program.
In order to compare the incidence rate of denosumab-associated severe hypocalcemia among patients with advanced CKD and ESRD (the target group of our integrated care) before and after the implementation of the integrated care program, the patients with advanced CKD and ESRD were divided into the following two groups: (1) those who were treated from January 2012 to December 2017 as the dataset of "before integrated care-control group "; and (2) those who were treated from January 2018 to December 2021 as the set of "after integrated care- intervention group ".
Comment:
The “integrated care program” appears to include (if not consistent primarily of) a calcium monitoring and supplementation program. It would appear that aspect of the program was at least consistent and could be evaluated against a control group. The aspects of the program “including consultations” are poorly explained, not standardized and their clinical relevance unknown.
Reply:
Yes. Thank you for your advice. As suggested, we added more descriptions about the consultations in the integrated osteoporosis care program paragraph of the Materials and Methods section. Osteoporosis-qualified nursing consultants (case managers) further referred patients with advanced CKD and ESRD to nephrologists for the evaluation of CKD-MBD and patient selection before denosumab treatment. According to the European consensus statement on the diagnosis and management of osteoporosis in advanced CKD and ESRD, CKD-MBD therapy should be optimized before denosumab treatment. The risks and benefits of denosumab treatment need to be balanced and discussed with the patient. Nephrologists are responsible for the optimizing of CKD-MBD therapy and the risks and benefits judgement before denosumab treatment. Denosumab may worsen the bone structure and vascular calcification among CKD patients with low turnover bone disease. We excluded patients with low parathyroid hormone and low bone metabolic markers from denosumab treatment.
(Please see the integrated osteoporosis care program paragraph of the Materials and Methods section in our clean copy of revised manuscript)
Comment:
The average day (with standard error) of hypocalcemia testing should be included. As the authors noted day 6 and 29 are within that 30 day window but may not be when hypocalcemia tends to occur. Nor is it clear that the level of hypocalcemia anticipated in constant during that period of time. Results of this study may be due to timing variation between groups.
Reply:
Yes. Thank you for your advice. The average day of hypocalcemia testing was 13.0 ± 7.2 days. As suggested, we mentioned the average day of hypocalcemia testing in the result section. Besides, we also mentioned the possible bias due to monitor timing variation between groups in the limitation paragraph of discussion section. patients with asymptomatic hypocalcemia could not be identified because laboratory monitoring was not undertaken, especially among patients treated before the integrated care program was implemented and patients without dialysis treatment. The frequency of serum calcium monitoring is significantly higher in dialysis patients than in patients without dialysis treatment, which may bias the episodes of hypocalcemia. The higher incidence of hypocalcemia among dialysis patients may be due to monitoring timing variation between groups in our study.
(Please see result and discussion section in our clean copy of revised manuscript)
Comment:
Sections that were not relevant to the study design were suggested to be omitted. The authors elected to add more on these unrelated topics which is counter to the recommendation.
Reply:
Yes. Thank you for your advice. As suggested, we reviewed our manuscript and removed sections that were not relevant to the study design.
(Please see our clean copy of revised manuscript)
Comment:
As requested, the discussion should evaluate the results in the context of the literature, specifically what evidence is there on the success of integrated care programs. What aspects of the program may/ or may not have contributed to the results (etc).
Reply:
Yes. Thank you for your advice. As suggested, we added more discussion about the approaches in our integrated care programs, including: (1) Vitamin D and calcium supplementation; (2) prophylactically increase the dialysate calcium concentration during denosumab treatment for dialysis patients; (3) warning system to increase the Doctors’ and healthcare professionals’ awareness.
(Please see discussion section in our clean copy of revised manuscript)
Comment:
The study has significant number of limitations and biases that must be delineated (many discussed by my fellow reviewers and myself). I would consider accepting if these limitations are appropriately added to the discussion.
Reply:
Yes. Thank you for your comment. As suggested, we address these limitations and biases in the limitation paragraph of discussion section. First, the retrospective nature of the study may have led to some unrecognized confounding factors to bias the findings. Second, some patients with asymptomatic hypocalcemia could not be identified because laboratory monitoring was not undertaken, especially among patients treated before the integrated care program was implemented and patients without dialysis treatment. The frequency of serum calcium monitoring is significantly higher in dialysis patients than in patients without dialysis treatment, which may bias the episodes of hypocalcemia. The higher incidence of hypocalcemia among dialysis patients may be due to monitoring timing variation between groups in our study. Third, the study population has a selection bias for one location. The case number of advanced CKD and ESRD in this study is relatively small because this study is a single-center retrospective study. Further studies with a larger sample size are required to confirm the efficacy of these approaches for reducing hypocalcemia complications.
(Please see the limitation paragraph of discussion section in our clean copy of revised manuscript)
Comment:
Revisions are in need of English spelling and grammar review.
Reply:
Yes. Thank you for your advice. As suggested, our revision had been checked the grammar and spelling by a native English speaker from MDPI Language Editing Service.
(Please see the English language editing certification in our supplementary material)

Reviewer 3 Report
No further comments.
Author Response
Thank you for your detailed review. We would also like to take this opportunity to express our thanks to you for your positive feedback and helpful comments for correction or modification.